# Genetic Basis of Follicle Development in Dazu Black Goat by Whole-Transcriptome Sequencing

**DOI:** 10.3390/ani11123536

**Published:** 2021-12-13

**Authors:** Lu Xu, Chengli Liu, Risu Na, Weiyi Zhang, Yongmeng He, Ying Yuan, Haoyuan Zhang, Yanguo Han, Yan Zeng, Weijiang Si, Xiao Wang, Chaonan Huang, Shiqi Zeng, Yongju Zhao, Zhongquan Zhao, Yongfu Huang, Guangxin E

**Affiliations:** Chongqing Key Laboratory of Forage & Herbivore, Chongqing Engineering Research Centre for Herbivores Resource Protection and Utilization, College of Animal Science and Technology, Southwest University, Chongqing 400716, China; lujiusym@163.com (L.X.); lcl222333@outlook.com (C.L.); narisu@swu.edu.cn (R.N.); Lwing0829@163.com (W.Z.); yongmenghe@163.com (Y.H.); 15703001957@163.com (Y.Y.); swuzhanghy@163.com (H.Z.); hyg2015@swu.edu.cn (Y.H.); zengyan@swu.edu.cn (Y.Z.); siweijiang666@163.com (W.S.); WX06210416@126.com (X.W.); hcn0911@163.com (C.H.); zsq13662056277@163.com (S.Z.); zyongju@163.com (Y.Z.); zhongquanzhao@126.com (Z.Z.); H67738337@swu.edu.cn (Y.H.)

**Keywords:** goat, noncoding RNA, competing endogenous RNAs, follicle development

## Abstract

**Simple Summary:**

The follicle development (FD) of a goat is precisely regulated by various noncoding RNAs (ncRNAs), especially by the regulatory mechanism of competing endogenous RNAs (ceRNAs). This study aimed to determine the expression patterns of messenger RNA (mRNA), long noncoding RNA, microRNA, and circular RNA during the FD of Dazhu black goats by whole-transcriptomic sequencing and analyze the regulatory mechanism of the ncRNA and ceRNA regulatory network. The results may lay a foundation for further research on FD and improving the reproductive performance of goats.

**Abstract:**

The follicle development (FD) is an important factor determining litter size in animals. Recent studies have found that noncoding RNAs (ncRNAs) play an important role in FD. In particular, the role of the regulatory mechanism of competing endogenous RNAs (ceRNAs) that drive FD has attracted increasing attention. Therefore, this study explored the genetic basis of goat FD by obtaining the complete follicular transcriptome of Dazu black goats at different developmental stages. Results revealed that 128 messenger RNAs (mRNAs), 4 long noncoding RNAs (lncRNAs), 49 microRNAs (miRNAs), and 290 circular RNAs (circRNAs) were significantly differentially expressed (DE) between large and small follicles. Moreover, DEmRNAs were enriched in many signaling pathways related to FD, as well as GO terms related to molecular binding and enzyme activity. Based on the analysis of the ceRNA network (CRN), 34 nodes (1 DElncRNAs, 10 DEcircRNAs, 14 DEmiRNAs, and 9 DEmRNAs) and 35 interactions (17 DEcircRNAs–DEmRNAs, 2 DElncRNAs–DEmiRNAs, and 16 DEmRNA–DEmiRNAs) implied that the CRN could be involved in the FD of goats. In conclusion, we described gene regulation by DERNAs and lncRNA/circRNA–miRNA–mRNA CRNs in the FD of goats. This study provided insights into the genetic basis of FD in precise transcriptional regulation.

## 1. Introduction

Goats (*Capra hircus*) have been economically important domesticated animals since the beginning of the agricultural civilization of mankind. Although female fertility is the most important economic performance indicator of goats, its genetic basis of molecular mechanisms remains unclear. Accordingly, follicle development (FD) is an important process that requires further research. Several studies have attempted to explain the genetic mechanism of FD in vertebrates [1,2]. However, the transcriptional regulation of ovarian follicles is highly complex and specific [3,4,5]. Although the transcriptional regulation of FD in mammals has been reported, its precise regulatory and developmental mechanisms has remain poorly understood.

Numerous noncoding RNAs have been confirmed to participate in known biological development processes by controlling protein-coding and noncoding genes [6,7,8]. In particular, the advent of next-generation, high-throughput technology has led to the identification of specific functional noncoding RNAs (ncRNAs) with roles in biological development and/or physiological responses [9,10,11]. Several studies have suggested that ncRNAs including microRNA (miRNAs), long noncoding RNA (lncRNAs), and circular RNA (circRNAs) play crucial roles in various biological processes in domestic animals [12,13,14]. Moreover, some studies have provided evidence supporting that lncRNA/circRNA–miRNA–messenger RNA (mRNA) interactions are important mechanisms underlying mammalian physiological development and diseases [15,16,17]. 

A series of studies on follicle and reproductive cell development has provided evidence of the interaction between mRNAs and ncRNAs. For example, miR-92b-3p negatively regulates *TSC1* in the mTOR/Rps6 signaling of primordial follicles in neonatal mouse ovaries [15], whereas miR-378 alters gene expression in cumulus cells and indirectly influences oocyte maturation competence in mice [18]. Moreover, lncRNA-H19 limits the number of follicles that mature, estradiol production, and ovulation concerning *AMH* in mice [19,20]. In the ovarian granulose cells (GC) of mice with polycystic ovary syndrome (PCOS), the lncRNA sequence read archive (SRA) inhibits proinflammatory cytokine production and NF-κB nuclear translocation induced by *DHEA*. This event subsequently alters insulin release and reduces ovarian damage and angiogenic factor production [10]. Competing endogenous RNAs (ceRNAs) are well known to bind competitively to miRNAs through their miRNA response elements to regulate the expression levels of miRNA target genes [21]. Some pieces of evidence supporting the possible participation of ceRNAs in many biological roles in vertebrates have been provided [22,23]. In particular, a series of ceRNA pairs has been implicated in FD. For example, circINHA promotes GC proliferation and inhibits GC apoptosis via *CTGF* by acting as a ceRNA that binds to miR-10a-5p [24]. *PWNR2* acts as a ceRNA to reduce the availability of miR-92b-3p for *TMEM120B* target binding during the maturation of PCOS oocytes, thereby playing an important role in the nuclear maturation of oocytes in PCOS [25]. Additionally, miR-486-5p has been inferred to play a key role in FD in PCOS by targeting *PRELID2*, whereas miR-4651 may be involved in inflammation by regulating its target gene [26]. All these results have indicated that the targeting of the regulatory effects of associated lncRNAs, circRNAs, miRNAs, and mRNAs may have a potential role in FD.

In the present study, we compared for the first time the expression levels of lncRNAs, mRNAs, miRNAs, and circRNAs in the large follicles (LFs) and small follicles (SFs) of Dazu black goats. Second, we constructed a preliminarily verified novel regulatory network mediated by lncRNA/circRNA–miRNA–mRNA ceRNA interactions to identify the key factors involved in FD. 

## 2. Materials and Methods

### 2.1. Ethics Statement and Sample Collection

The experimental conditions of this study were approved by the Committee on the Ethics of Animal Experiments of Southwest University (No. [2007] 3) and the Animal Protection Law of China. The ovaries of four Dazu black goats approximately 1.5–2 years of age were collected. The follicles were stripped in physiological saline at 37 °C. The first six largest follicles of the bilateral ovaries of each animal were separated and sorted by diameter. The follicles from all animals were clustered into groups 1 to 6 based on size (GF_1, GF_2, GF_3, GF_4, GF_5, and GF_6). The three groups with the largest follicle size were considered as the LF cluster (diameter > 4 mm, GF_C (GF_1, GF_2 and GF_3), control group), and the other three groups with the smallest follicle sizes were designated as the SF cluster (diameter < 4 mm, GF_T (GF_4, GF_5, and GF_6), experimental group) in accordance with a previous study on Dazu black goats (China National Commission of Animal Genetic Resources, 2011) [27]. 

### 2.2. Isolation of Total RNA and Transcriptome Sequencing

Total RNA (for mRNAs, lncRNA, and circRNAs) was extracted from each follicle-size group by using TRIzol^®^ Reagent following the manufacturer’s protocol (Invitrogen, Waltham, MA, USA). Genomic DNA was removed using DNase I (TaKara, Japan). RNA quality was determined with ND-2000 (NanoDrop Technologies, Wilmington, DE, USA). Equal amounts of RNA from each follicle-size group were selected for library preparation. Ribosomal RNA was removed using Epicentre Ribo-zero Ribosomal RNA (rRNA) Removal Kits (Epicentre, Madison, WI, USA), and the rRNA-free residue was precipitated with ethanol. High-strand-specificity libraries were generated with NEBNext Ultra Directional RNA Library Prep Kit for Illumina (NEB, Ipswich, MA, USA). DNA 1000 Assay Kit (Agilent Technologies, Santa Clara, CA, USA) or High-Sensitivity DNA assay Kit (Agilent Technologies, Santa Clara, CA, USA) was used for library quality inspection. Finally, ABI StepOnePlus Real-Time PCR System (Life Technologies, Carlsbad, CA, USA) was used for quantitative and pooling analyses, and transcriptome sequencing was conducted using Hiseq2500 with PE150.

Total RNA (miRNA) was extracted with polyethylene glycol 8000, and RNA molecules within a size range of 18–30 nt were enriched through polyacrylamide gel electrophoresis. Then, 3′ adapters were added, and 36–44 nt RNAs were enriched. The RNAs were also ligated with 5′ adapters. The ligation products were reverse transcribed by PCR amplification, and PCR products 140–160 bp in size were enriched to generate a cDNA library. Finally, sequencing was performed using an Illumina HiSeq 2500 platform (2 × 125 bp read length, Guangzhou, China).

### 2.3. Quality Analysis, Mapping, Transcript Assembly, and Coding RNA Identification

Clean data were obtained by removing low-quality reads and adapter sequences by utilizing SeqPrep (https://github.com/jstjohn/SeqPrep accessed on 24 October 2019) and Sickle (https://github.com/najoshi/sickle accessed on 27 October 2019) with the default parameters. The Q20, Q30, and GC contents of clean data were calculated, and all subsequent analyses were based on high-quality data. The clean reads of each group were aligned to goat genome (ARS1, GCA_001704415.1) by using Bowtie v2.0.6 [28] and TopHat v2.0.9 [29] software, respectively. Transcript assembly and abundance estimation were performed with TopHat [29] and Cufflinks [30]. BLASTX was used to search the NR and KEGG databases (E-value < 10^−5^).

All reconstructed transcripts were compared with the reference genome to identify novel gene transcripts and classified with Cuff compare. Genes with the class code “uijxceo (transcript unknown or in the intergenic region)” were defined as novel genes. Then, the following parameters were used to determine a reliable novel gene: transcript length greater than 200 bp and number of exons greater than 2. Subsequently, the novel gene was compared with genes in the NR and KEGG databases to obtain protein function annotations. Gene-expression level was normalized by the fragments per kilobase of transcript per million mapped reads (FPKM) method.

### 2.4. LncRNA Identification

Four types of software, namely, Coding-Noncoding-Index v2 [31], CPC v0.9-r2 (https://cpc.cbi.pku.edu.cn/ accessed on 14 November 2019) [32], PhyloCSF [33], and Pfam [34], were used to assess the protein-coding potential of the novel transcripts with default parameters. The intersections of non-protein-coding potential results were selected as lncRNAs. The transcripts that passed through all these stages were considered to be lncRNAs. LncRNA expression levels were reflected as FPKM.

### 2.5. MiRNA Identification and Target-Gene Prediction

The GeneBank database (release 209.0) and Rfam database (11.0) were used to annotate data and maximize the removal of rRNAs, small cytoplasmic RNAs (scRNAs), small nucleolar RNAs (snoRNAs), small nuclear RNAs (snRNAs), and transfer RNAs (tRNAs) from small RNAs (sRNAs). The genomic origin of the tag sequence was identified by comparing the tag sequence with the reference genome through Bowtie (v1.1.2) with the parameter settings -v 0 --best --strata –a, where v means the maximum allowable number of mismatches. The repetitive sequence region was identified using Repeat Masker version open-4.0.6 (parameter settings: -engine wublast -s -no_is -cutoff 255 -frag 20 000). Then, the sRNA tag sequence of the repeat associate was identified based on the result of genome alignment and the position of the repeated sequence on the genome. All clean tags were aligned with the reference genome. Tags that were mapped to repeat sequences were also removed.

The tags were compared with miRNA precursors and mature bodies by utilizing the miRBase database (release 21) (https://www.MiRbase.org/ accessed on 3 December 2019) to obtain known miRNAs. Then, all unannotated tags were aligned with the reference genome, and the novel miRNA candidates were identified using default parameters in accordance with their genomic positions and hairpin structures predicted with Mireap_v0.2. MiRNA sequences. Family information was obtained from TargetScan website (https://www.targetscan.org/ accessed on 12 December 2019). The miRNA (existing, known, and novel miRNAs) expression levels in each group were calculated and normalized to transcripts per million (TPM) with the following formula: TPM = Actual miRNA counts/(total counts of clean tags × 106).

We predicted the target relationship between DEmiRNAs and DEmRNAs with default parameters by using RnaHybrid (v2.1.2) + svm_light (v6.01), MIRANDA (v3.3a), and TargetScan (v7.0). The intersections of the results had high credibility for selection as the predicted target genes of miRNA. Expression correlation between mRNA and miRNA was evaluated using the Spearman rank correlation coefficient (SCC). Pairs with SCC < −0.7 were selected as negatively coexpressed mRNA–miRNA pairs wherein mRNA was DEmiRNA-TGs.

### 2.6. CircRNA Identification and Source-Gene Prediction

After aligning the clean reads to the goat reference genome, the junctions of the unmapped reads were identified using a back-splice algorithm and combining all samples’ comparison results. The prediction results for circRNAs were visualized using Findcirc software [34] with the following criteria: (a) Breakpoint = 1, and circRNAs with one clear breakpoint were retained; (b) Anchor_overlap ≤ 2, and the two anchor reads of each read aligned to the genome position overlap could not exceed 2 bp; (c) Edit ≤ 2, and only 2 bp mismatches were allowed; (d) n_uniq > 2, and unique reads must be greater than 2; (e) best_qual_A > 35 or best_qual_B > 35, and the highest mapping result of the single anchor read of each read must be higher than that of the second-ranked result with a score of at least 35; (f) n uniq > int(samples/2), and the unique reads supporting the circRNAs must be greater than half of the total number of samples; (g) The length of the circRNAs was less than 100 kb. The back-spliced reads per million (RPM) mapped reads of back-spliced junction reads was used to quantify the expression abundance of circRNAs with the following formula: RPM = 10^6^ C/N, where C is the only back-spliced junction read that is mapped to a circRNA, and N is the total number of back-spliced junction reads. The RPM method can eliminate the influence of different sequencing-data volumes on the calculation of circRNA expression. Therefore, the calculated expression can be directly used to determine differential expression between groups. Furthermore, DEcircRNA-SGs were obtained using Findcirc software [34].

### 2.7. Differentially Expressed (DE) RNA Identification and Enrichment Analysis

DE RNAs between different follicle size groups were identified using the edgeR package (https://www.bioconductor.org/packages/release/bioc/html/edgeR.html accessed on 4 January 2020) with the following criteria: DElncRNAs and DEmRNAs with |log2FC| > 1 and false discovery rate < 0.05; and DEcircRNAs and DEmiRNAs with |log2FC| > 1 and *p* < 0.05. Furthermore, GO functional enrichment and KEGG pathway analysis were performed using Goatools (https://github.com/tanghaibao/Goatools accessed on 11 January 2020) and KOBAS (https://kobas.cbi.pku.edu.cn/genelist/ accessed on 14 January 2020) [35]. DEmRNAs, DEmiRNA-TGs, and DEcircRNA-SGs were considered to be significantly enriched in GO terms and metabolic pathways when their corrected *p*-value was less than 0.05.

### 2.8. CeRNA Regulatory Network (CRN) Construction

The CRN was constructed as follows: (a) The expression correlation between DEmRNAs–DEmiRNAs, DElncRNAs–DEmiRNAs, or DEcircRNAs–DEmiRNAs was evaluated using the SCC. Pairs with SCC < −0.7 were selected as negatively coexpressed circRNA–miRNA pairs, lncRNA–miRNA pairs, or mRNA–miRNA pairs, wherein mRNA, circRNA, and lncRNA were miRNA target genes, and all RNAs were DE. (2) The expression correlation between circRNA/lncRNA–mRNA was evaluated using the Pearson correlation coefficient (PCC). Pairs with PCC > 0.9 were selected as coexpressed circRNA/lncRNA–mRNA pairs, wherein mRNA and lncRNA/circRNA were targeted and negatively coexpressed with a common miRNA. (3) As a result, only gene pairs with *P* values less than 0.05 were selected. The lncRNA/circRNA–miRNA–mRNA network was constructed and visualized using Cytoscape software (v3.6.0) [36].

### 2.9. Reverse-Transcription Quantitative PCR (RT-qPCR) Verification

The samples used in the qPCR analyses were the same as those used in the RNA-seq study. The cDNA synthesis of mRNA, lncRNA, and circRNA was performed with PrimeScript™ RT reagent kit with gDNA Eraser (TaKaRa, Japan). The reverse transcription of miRNA was performed using Mir-X™ miRNA First-Strand Synthesis (TaKaRa, Japan), and the primers of all RNAs are shown in Appendix A. QPCR of all RNA was performed with 10 μL of TB Green Premix Ex Taq II (Tli RNaseH Plus,) (TaKaRa, Japan), 7.0 μL of H_2_O, 0.5 μL of each primer (10 pmol/mL), and 2.0 μL of cDNA (< 100 ng). According to the manufacturer’s instructions: 95 °C for 10 min for 1 cycle, followed by 40 cycles of 95 °C for 15 s and 60 °C for 45 s. The reaction was performed on an Applied Biosystems StepOnePlusTM Real-Time PCR System (Life Technologies, Carlsbad, CA, USA). Melting curves were constructed to verify that only a single PCR product was amplified. Within runs, the samples were assayed in triplicate, with standard deviations of the threshold cycle values not exceeding 0.5; each QPCR run was repeated at least three times. Negative (without template) reactions were performed within each assay. We used the 2^−ΔΔCt^ method for relative quantitation between samples, and the cycle-threshold cycle values were normalized to housekeeping genes (*GAPDH* and U6). Significant differences were determined by ANOVA.

## 3. Results

According to the previous study’s results on the determination of follicular development in Dazu black goat by B-ultrasound technology from our lab, the diameter of Graafian follicle was defined as greater than 4 mm [27]. Therefore, the diameter length of follicles larger than 4 mm and smaller than 4 mm was divided into GF_T and GF_C, respectively. The follicle groups GF_1 to GF_6 were defined by decreasing their diameter.

A total of six lncRNA libraries and six miRNA libraries were sequenced on the Illumina HiSeqTM 2500 platform and subjected to preliminary filtering, obtaining about a total of 97,020,363,900 bp clean data of lncRNA sequence and 84,485,902 clean reads of small sequence. The data were uploaded to the SRA database, with project accession numbers PRJNA722567 and PRJNA715900. The valid ratio of each library is shown in Table 1 and Table 2. Details are shown in Appendix A.

### 3.1. DEmRNAs, DElncRNAs, and Functional Annotation

A total of 25 760 mRNAs were obtained in this study, and 128 significant DEmRNAs were identified between the GF_C and GF_T groups, including 63 upregulated and 65 downregulated GF_T compared with GF_C. Additionally, four significant DElncRNAs were identified from 4095 scanned lncRNAs. A heatmap of hierarchical clustering of DEmRNAs or DElncRNAs was further generated to visualize the overall pattern of gene expression (Figure 1A,B). All DEmRNAs and lncRNAs are shown in Appendix A. KEGG results indicated that 41 of the 128 genes were annotated in 102 KEGG pathways. Most of the KEGG pathways involved signal transduction of environmental information processing (e.g., the Wnt signaling pathway, MAPK signaling pathway, TGF-β signaling pathway, Hippo signaling pathway, and PI3K-Akt signaling pathway). The top 20 signaling pathways are shown in Figure 1C and Appendix A. GO annotation analysis results of 128 mRNAs revealed that 1079 GO terms were enriched. They were related mostly to cell composition and molecular binding, e.g., protein binding, nucleus, cytosol, cytoplasm, Rho GTPase binding, ATP binding, and calmodulin binding (Figure 1D and Appendix A).

### 3.2. DEmiRNAs and Functional Annotation of DEmiRNA-TGs

A total of 49 significant DEmiRNAs from a total 1110 miRNA were identified between the GF_C and GF_T groups; 25 were upregulated and 24 were downregulated (Figure 2A and Appendix A). Meanwhile, a total of 60 negatively correlated targeting relationship pairs were predicted using DEmiRNAs and DEmRNAs, including 28 DEmiRNAs and 32 DEmRNAs. The DEmiRNA-TGs were primarily enriched in the GO terms of apical junction assembly, nucleus, histone H3-K4 methylation, apoptotic process, RNA binding, and protein binding (Figure 2B and Appendix A). The enriched KEGG pathways included Cushing syndrome, the Wnt signaling pathway, the MAPK signaling pathway, and the PI3K-Akt signaling pathway (Figure 2C and Appendix A). Moreover, miR-383 and miR-135b-5p were found to be the key miRNAs with a functional analysis of DEmiRNAs, and their target genes and the main signal pathways of target-gene enrichment are shown in Figure 2D.

### 3.3. DEcircRNAs and Functional Annotation of DEcircRNA-SGs

A total of 290 significant DEcircRNAs in 13,583 circRNAs (Appendix A), including 155 upregulated and 135 downregulated, were identified from GF_T compared with GF_C (Figure 3A and Appendix A). Furthermore, we identified 290 DEcircRNAs, which originated from 259 source genes. GO analysis revealed that a total of 1986 GO terms were annotated, including 191 terms that were significantly enriched (e.g., protein binding, GTPase activator activity, cytosol, and ATP binding; Figure 3B and Appendix A). Moreover, 84 DEcircRNA-SGs were enriched in 192 pathways. In particular, most pathways from the top 20 enriched pathways originated from the signal-transduction class, including the TNF signaling pathway, TGF-β signaling pathway, FoxO signaling pathway, ErbB signaling pathway, and phosphatidylinositol signaling system (Figure 3C and Appendix A).

### 3.4. Identification of CRN and Functional Annotation

As shown in the CRN analysis results, a total of 34 nodes (1 DElncRNAs, 10 DEcircRNAs, 14 DEmiRNAs, and 9 DEmRNAs) and 35 interaction relationships (17 DEcircRNA-DEmiRNA, 2 DElncRNA-DEmiRNA, and 16 DEmRNA-DEmiRNA) were identified (Figure 4 and Appendix A). Nine coding genes (TCONS_00000537 (*PAK2*), TCONS_00013694 (*TMTC2*), TCONS_00027577 (*FAXC*), TCONS_00043904 (*KMT2A*), TCONS_00054440 (*NUFIP2*), XM_018059280.1 (*QSER1*), XM_018060043.1 (*DSTYK*), XM_018061337.1 (*PITPNM2*), and XM_018063778.1 (*FBF1*)) were verified in the final CRN. Moreover, the result of GO annotation analysis of mRNA in CRN revealed that *FBF1* was enriched in 108 GO terms, including 68 GO terms that were significantly different (apical junction assembly, regulation of histone H3-K9 acetylation, epithelial cell migration, phosphatidylinositol transfer activity, and unmethylated CpG binding). The KEGG result implied that *KMT2A* was enriched in the lysine degradation signal pathway. Cushing syndrome, transcriptional misregulation in cancer, and TCONS_00000537 (*PAK 2*) were annotated in three pathways (e.g., salmonella infection, yersinia infection, and PI3K-Akt signaling pathway, respectively). Furthermore, novel_circ_005369 chi-miR-10a-5p, and XM_018061337.1 (*PITPNM2*) had the highest degree of connectivity (Appendix A). In the CRN, nodes with high connectivity often had important biological significance, and these genes were considered as hub genes.

### 3.5. Validation of DE RNAs by RT-qPCR

We randomly selected a total of 16 RNAs (mRNA, lncRNA, miRNA, and circRNA) to confirm the accuracy of the RNA-seq results in this study, which were significantly and DE in the RNA-seq results for RT-qPCR verification. Results showed that three mRNAs, one lncRNA, four miRNAs, and eight circRNAs were significantly and DE between the GF_T and GF_C groups by RNA-seq (B,D,F). However, only the relative expression levels (REL) of four miRNAs were significantly and DE by QPCR (Figure 5C; *p* < 0.05). The REL of other RNAs (mRNA, lncRNA, and circRNA) showed a consistent trend compared with the RNA-seq results (Figure 5A,E). Therefore, the sequencing and analysis results of the transcriptome were reliable.

## 4. Discussion

Considerable attention has been paid to the pathways involved in the regulation of ovarian function, such as corpus luteum formation and steroid production [37,38,39]. In this study, numerous DEmRNAs were enriched in Wnt [40], MAPK [41], TGF-β [42], PI3K-Akt [43], Hippo [44], and glutathione metabolic [45] signaling pathways, which have been verified to be associated with FD. Such pathways could improve the treatment of infertility with reduced ovarian reserve in the future [43,46,47,48]. Meanwhile, these pathways could inhibit GC autophagy, and exogenous toxic substances could hinder the growth of ovarian GCs [49,50].

In particular, the PI3K-Akt signaling pathways are involved in the maintenance of ovarian function, FD, premature ovarian failure, and ovarian activation, which can activate dormant follicles in vitro for a short period of time and generate numerous mature germ cells [51,52,53]. In the present study, we found that *LAMA5*, as a DEmRNA, was enriched in the PI3K-Akt pathway and played a key role in the development of endometrial and epithelial tissue in pregnant women [49,54]. Meanwhile, the *SPP1* gene can regulate the proliferation of cumulus cells around oocytes through the PI3K-Akt pathway [48]. Although PI3K-Akt had no confirmed relations with FD and the reproduction of goats, many studies have found that the PI3K-Akt pathway was closely related to FD in pig and some bovine animals [55,56].

Moreover, some studies have shown that the Hippo signaling pathway also plays an important role in regulating oocyte polarity, oocyte cavity structure, and ovarian-function stability [57,58,59,60]. Herein, DEmRNA *ACP* was enriched in the Wnt and Hippo signaling pathways, indicating that the *ACP* gene may be involved in FD through multiple signaling pathways. Notably, DEmRNA *SMURF2* (TCONS_00055607) was enriched in the TGF-β signaling pathway, which is extensively recognized to participate in FD [61,62,63], particularly through synergistic involvement with bone morphogenesis protein and growth factors [64,65,66]. Therefore, *SMURF2* and *ACP* genes may be involved in the molecular regulation of FD in goats.

Numerous studies have shown that a series of miRNAs plays an important role in GCs, FD, and atresia disorders in the ovaries [67,68]. For example, miR-128 was confirmed to be involved in follicle selection by lipid regulation in ovule development [69]. Meanwhile, a series of studies has demonstrated that miR-449a is closely related to ovarian function and FD in humans [70] and cattle [71]. The expression of miR-10a-5p was also the highest in all DEmiRNAs of this study. Furthermore, miR-10a-5p could regulate the apoptosis of follicular GCs in pigs [24]. In the current work, miR-128, miR-383, miR-10a-5p, and miR-449a-5p were significant DEmiRNA between groups, suggesting that they may also be involved in the FD of goats.

Interestingly, a previous study has shown that miR-383 is the differential expression between DFs and SFs of Dazu black goats, and the expression shows a downward trend with an increased follicle diameter. It may play an inhibitory role in FD [70]. Meanwhile, miR-383 presents a DE pattern in GCs of bovine atretic follicles and dominant follicles [71,72], and inhibits the translation of *RBMS1* by affecting the mRNA stability of *RBMS1*, thereby enhancing the release of estradiol in ovarian GCs and mediating steroid production [73,74]. Here, we mapped the regulatory network of miR-383 (Figure 2A) and found that the miR-383 may regulate FD through the Wnt, PI3K-Akt, and Hippo signaling pathways [48,75]. Another significant DEmiRNA (miR-135b-5p) between groups also caught our attention. A previous report has indicated that miR-135b-5p can inhibit *HIF1AN* expression and improve the proliferation ability of ovarian cancer cells [76]. Furthermore, the potential target gene (*APC*) of miR-135b-5p is closely related to the meiosis of mouse oocytes [77].

The present study also found numerous DEcircRNAs, suggesting that circRNAs may be closely related to FD in goats. Specifically, novel_circ_005179 was expressed only in the GF_C group, whereas novel_circ_011344 was expressed only in GF_T, indicating that these circRNAs may play an important role in the FD of goats. Notably, novel_circ_005179 had the largest expression divergence between groups, and its source gene was *AKAP2*, which was related to signaling pathways involved in mediating the cessation and recovery of meiosis [78]. Additionally, the novel_circ_012471 was greatly expressed in the LF group, and its source gene (*RBPMS2*) was confirmed to be related to oocyte development in zebrafish [79].

To date, numerous studies have found that the regulatory model of ceRNA is involved in the reproductive process of human and animals. For example, circLDLR in exosomes serves as a vital mediator to regulate estradiol secretion by sponging miR-1294 to repress the *CYP19A1* gene in humans [80]. LncRNA-MALAT1 regulates the GCs of mouse apoptosis and 17β-estradiol synthesis by regulating the miR-205/CREB1 axis [81]. CircINHA could resist GCs apoptosis by upregulating *CTGF* as a ceRNA of miR-10a-5p in pig ovarian follicles [24]. All above-mentioned studies indicate that the regulatory mechanism of ceRNA plays an important role in the development of mammalian follicles.

Notably, the results of the current study showed that XR_001918824.1 and novel_circ_009670 could regulate the expression of *NUFIP2* and *FBF1* genes by regulating miR-10a-5p. Meanwhile, the lncRNA-XR_001918824.1 and circRNA-Novel_circ_009670 in this network were expressed only in SFs. Specifically, the expression level of miR-10a-5p was the highest in DEmiRNAs, and it was one of the miRNAs with the highest connectivity in the CRN. The ability of its target gene (*FBF1*) to regulate the meiosis of mouse oocytes was confirmed [82]. Thus, miR-10a-5p may regulate FD development in goats through the CNR. Finally, the nine coding RNAs (*PAK2*, *TMTC2*, *FAXC*, *KMT2A*, *NUFIP2*, *QSER1*, *DSTYK*, *PitPNM2*, and *FBF1*) obtained by CRN analysis in this study were found to be closely associated with animal germline. For example, a series of studies has shown that *DSTYK* is involved in the conduction of the mTORC1/TFEB and ERK1/2 signaling pathways [83,84], and both pathways are closely related to the development of ovarian follicles and ovulation [85,86]. Furthermore, the *PAK2* and *KMT2A* genes are related to the FD of vertebrates and the proliferation and differentiation of ovarian GCs [87,88]. The *PAK2* gene also plays a role in the proliferation of chicken ovarian GCs through the RAFS/ERK MAPK pathway [88]. Therefore, these results suggest that these genes may participate in the FD of goats through CRN regulation.

## 5. Conclusions

We obtained numerous key genes and signaling pathways related to the FD of goats. A series of ceRNAs networks was found to indicate that they may be involved in FD. Therefore, this study provided a new understanding of the genetic basis of goat FD. Further in-depth investigations are necessary to validate the ceRNA-regulated mechanisms of ncRNAs and coding RNAs.

## Figures and Tables

**Figure 1 animals-11-03536-f001:**
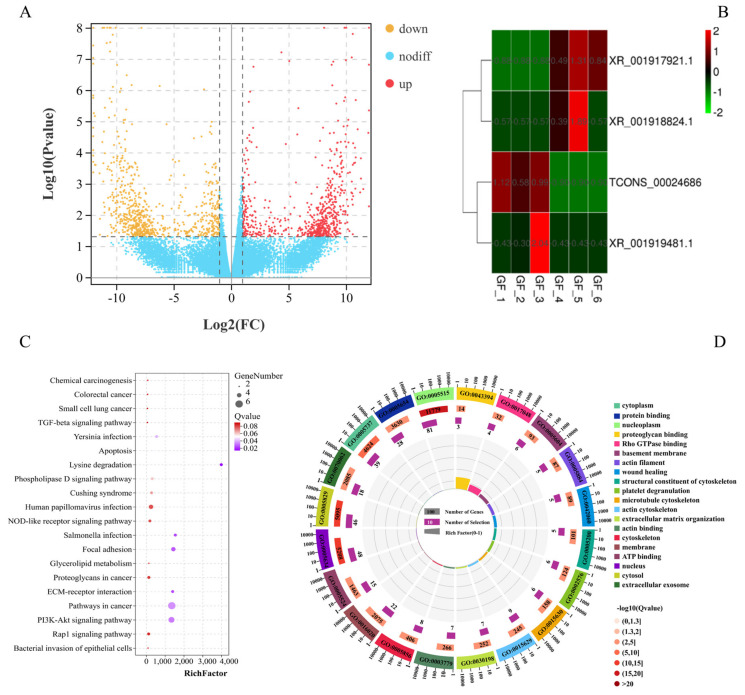
Functional analysis of DEmRNAs and DElncRNAs between the LFs and SFs of Dazu black goat: (**A**), volcano plot of 128 DEmRNAs; red represents up-regulated expression and orange represents down-regulated expression; (**B**), clustering heat map of 4 DElncRNAs; red represents up-regulated expression and green represents down-regulated expression; (**C**), top 20 pathways of KEGG enrichment analysis; and (**D**), top 20 terms of GO enrichment analysis.

**Figure 2 animals-11-03536-f002:**
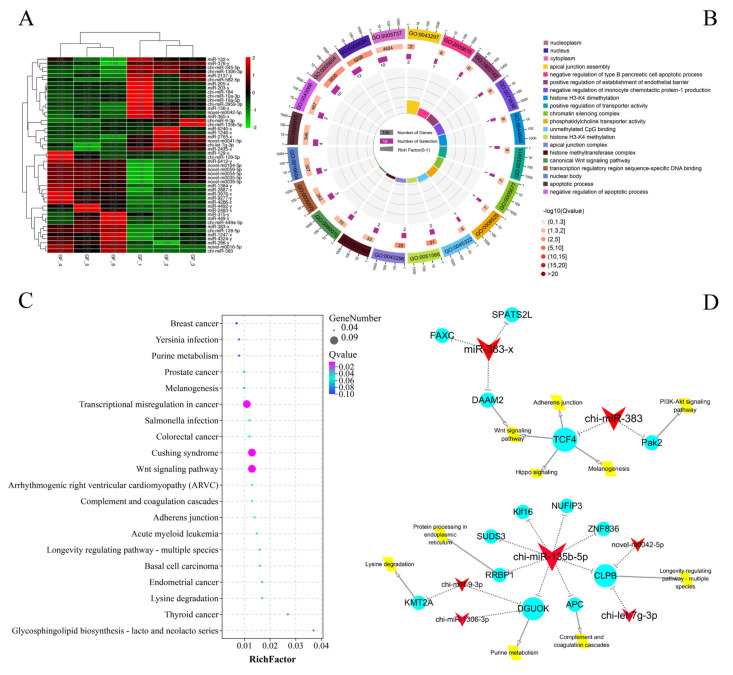
Functional analysis DEmiRNAs between LFs and SFs of Dazu black goats: (**A**), clustering heat map of 49 DEmiRNAs; red represents up-regulated expression and green represents down-regulated expression; (**B**), top 20 terms of GO enrichment analysis; (**C**), top 20 pathways of KEGG enrichment analysis; and (**D**), key miRNA regulation network in DEmiRNAs between LFs and SFs of Dazu black goats; red represents DEmiRNAs, blue represents DEmRNAs, and yellow represents the major pathways; the size represents the number of DEmiRNA-TGs, i.e., a larger size means greater number of DEmiRNA-TGs.

**Figure 3 animals-11-03536-f003:**
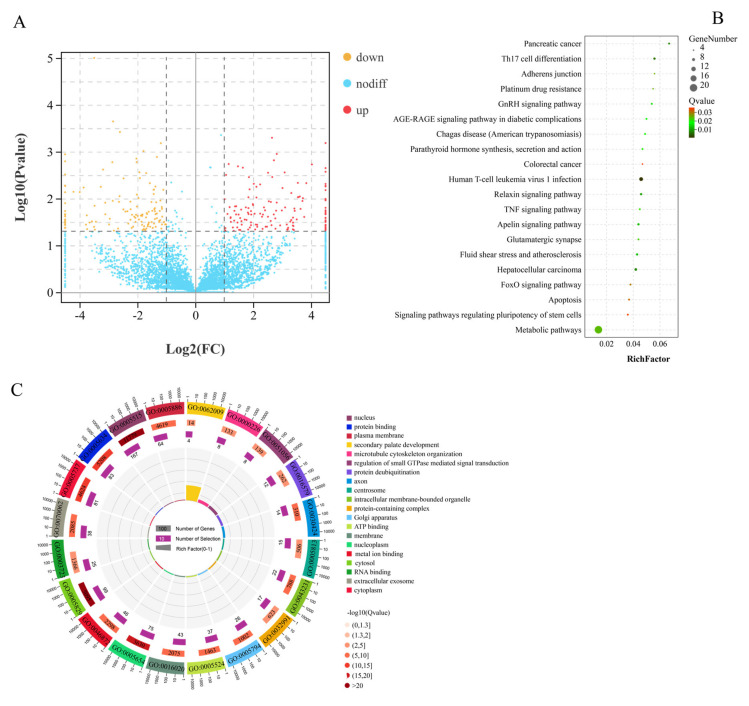
Functional analysis of DEcircRNAs between LFs and SFs of Dazu black goats: (**A**), volcano plot of 290 DEcircRNAs; red represents up-regulated expression and orange represents down-regulated expression; (**B**), top 20 pathways of KEGG enrichment analysis; and (**C**), top 20 terms of GO enrichment analysis.

**Figure 4 animals-11-03536-f004:**
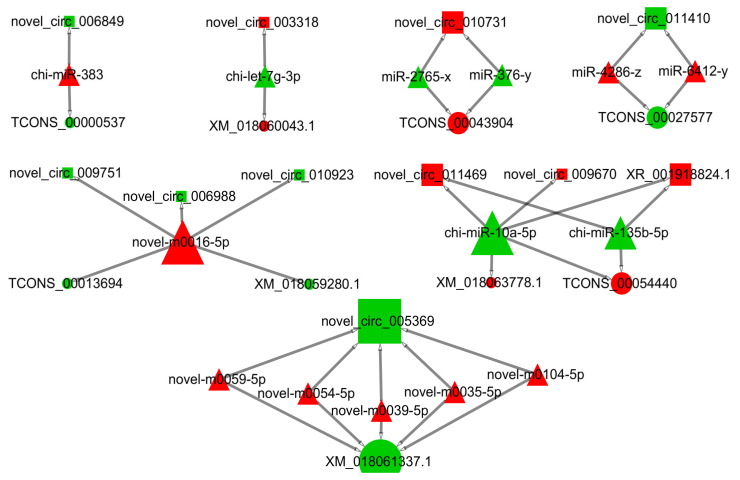
LncRNA/circRNA–miRNA–mRNA ceRNA network. Rectangles indicate lncRNA or circRNA, triangles represent miRNAs, and circles represent mRNAs. Red indicates up-regulation, and green indicates down-regulation.

**Figure 5 animals-11-03536-f005:**
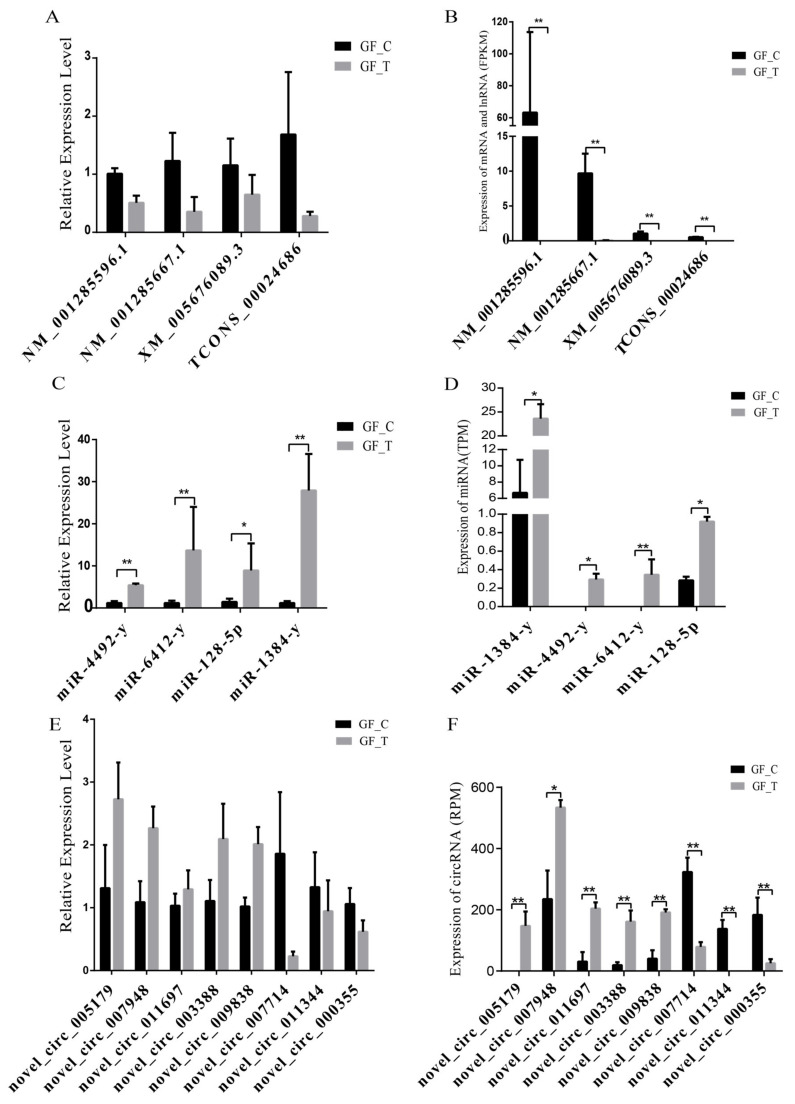
DEmRNAs, DElncRNAs, DEmiRNAs, and DEcircRNAs verified by RT-qPCR in LFs and SFs RNA-seq results: A, REL of mRNA and lncRNA by RT-qPCR; B, expression levels of mRNA and lncRNA (FPKM) by RNA-seq; C, REL of miRNA by RT-qPCR; D, expression levels of miRNA (TPM) by RNA-seq; E, REL of circRNA by RT-qPCR; and F, expression levels of circRNA (RPM) by RNA-seq; *: *p* < 0.05; **: *p* < 0.01.

**Table 1 animals-11-03536-t001:** Summary statistics of lncRNA and mRNA sequencing data.

Sample	Clean Data (bp)	HQ Clean Data (bp)	Q30	GC
GF_1	15375727800	14965534035 (97.33%)	14372884127 (96.04%)	6637373901 (44.35%)
GF_2	17141768700	14965534035 (97.12%)	15943146337 (95.77%)	7477704890 (44.92%)
GF_3	12268004700	14965534035 (96.99%)	11454639659 (96.26%)	5280700810 (44.38%)
GF_4	17644711200	14965534035 (97.12%)	16443803331 (95.87%)	7735202025 (45.10%)
GF_5	17372614800	14965534035 (97.27%)	16214206723 (95.95%)	7518644939 (44.49%)
GF_6	17217536700	14965534035 (97.17%)	16041939269 (95.89%)	7570524208 (45.25%)

**Table 2 animals-11-03536-t002:** Summary statistics of miRNA sequencing data.

Sample	Clean_Reads	HQ Clean_Reads	3′Adapter_Null	5′Adapter
GF_1	11,227,206 (100%)	11,075,835 (98.65%)	57,261 (0.5170%)	4470 (0.0404%)
GF_2	10,973,564 (100%)	10,829,968 (98.69%)	88,118 (0.8136%)	6082 (0.0562%)
GF_3	10,919,757 (100%)	10,775,502 (98.68%)	62,409 (0.5792%)	7034 (0.0653%)
GF_4	13,403,182 (100%)	13,219,081 (98.63%)	76,083 (0.5756%)	10,947 (0.0828%)
GF_5	11,697,146 (100%)	11,540,707 (98.66%)	67,517 (0.5850%)	8372 (0.0725%)
GF_6	14,195,632 (100%)	14,004,736 (98.66%)	70,896 (0.5062%)	6492 (0.0464%)

## Data Availability

All data was generated and uploaded to the NCBI database. (PRJNA715900 and PRJNA722567).

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
