# Peer review of "Genetic Basis of Follicle Development in Dazu Black Goat by Whole-Transcriptome Sequencing"

_animals, 2021, doi:10.3390/ani11123536_

Round 1
Reviewer 1 Report
The manuscript provides a list of RNAs associated with follicle developments at different stages. Follicles were collected from 4 animals. Proper sample preparation techniques and wide range of software have been applied to find differentially expressed RNAs. Validation of randomly selected RNAs from the top hits reflected, that a certain ration of these hits are indeed differentially expressed.
I suggest the MS for publication.
There are a few notices pointing on some places to be remedied:
Line 12:
The shortening of ‘follicular development’ and ‘competing endogenous RNA’ are given in brackets. Please do the same with the abbreviations found in the text: ln, lnc, nc, mi, circ, s, sn, sno, and DE.
However these abbreviations are resolved in Table 3., under the ‘Abbreviations’ but I have not found any reference to Table 3. in the text.
Line 97, 103:
‘NanoDrop Technologies’,
‘Agilent Technologies’: Country is missing.
Line 122:
It might be better: ‘(hits with E value lower than 10-5 were accepted)’
Line 327:
’…consistent trend…’
How this trend will be turned to be significant later on? Answer should go into the discussion.
Line 138:
’ Therefore, the…’
This sentence should go into the discussion.
Author Response
Reply to Reviewer: 1
The manuscript provides a list of RNAs associated with follicle developments at different stages. Follicles were collected from 4 animals. Proper sample preparation techniques and wide range of software have been applied to find differentially expressed RNAs. Validation of randomly selected RNAs from the top hits reflected, that a certain ration of these hits are indeed differentially expressed.
I suggest the MS for publication.
There are a few notices pointing on some places to be remedied:
Answer: Thanks your suggestions and we had done the revised as following:
Question 1:Line 12:The shortening of ‘follicular development’ and ‘competing endogenous RNA’ are given in brackets. Please do the same with the abbreviations found in the text: ln, lnc, nc, mi, circ, s, sn, sno, and DE.
Answer 1: Thanks your suggestions and we had done the revised as following:
non-coding RNAs (ncRNA), messenger RNA (mRNA), Long non-coding RNA (lnRNA), microRNA (miRNA) and circular RNA (circRNA), differentially expressed (DE), ribosomal RNA (rRNA), rRNAs, small cytoplasmic RNAs (scRNAs), small nucleolar RNAs (snoRNAs), small nuclear RNAs (snRNAs), and transfer RNAs (tRNAs) from the small RNAs (sRNAs).
Question 2: However these abbreviations are resolved in Table 3., under the ‘Abbreviations’ but I have not found any reference to Table 3. in the text.
Answer 2: Thanks your suggestions and Considering that these abbreviations and full names have already been given in the article, so we have removed Table 3.
Question 3: Line 97, 103:‘NanoDrop Technologies’, ‘Agilent Technologies’: Country is missing.
Answer 3:ND-2000 (NanoDrop Technologies, USA), DNA 1000 Assay Kit (Agilent Technologies, USA)
Question 4: Line 122: It might be better: ‘(hits with E value lower than 10-5 were accepted)’
Answer 4: Thanks your suggestions and we had done the revised as following: (E-value < 10−5)
Question 5: Line 327:’…consistent trend…’How this trend will be turned to be significant later on? Answer should go into the discussion.
Answer 5: It may play an inhibitory role in Follicular development.
Question 6: Line 138: Therefore, the…’This sentence should go into the discussion.
Answer 6: This section has been deleted.

Reviewer 2 Report
In the title: Investigated should be Investigating; choose either "using" or "by" not both;
L151: This sentence is unclear, specifically mature bodies of goats. Do you mean mature microRNAs? If so please clarify.
L228: Results
Please include more details about the follicle groups and descriptive characteristics on the follicles utilized for this study. It is important for the reader to understand the differences present between the GF groups. For example is the GF_1 group >5m while GF_6 <3 or something similar? What was the median follicle size for each of the comparison groups? The characteristics of the follicles used are results and should be included.
Author Response
Reply to reviewer’s Comments
Dear Editor and reviewers
Thanks your hard work and help!
We had done the modification as your suggestion. Meanwhile, the grammar of the manuscript has been polished by the service of EssayStar. The revisions are highlighted in red in the manuscript.
Please check as following:
Reply to Reviewer: 2
Question 1: In the title: Investigated should be Investigating; choose either "using" or "by" not both;
Answer 1:Genetic Basis of Follicle Development in Dazu Black Goat by Whole-Transcriptome Sequencing
Question 2: L151: This sentence is unclear, specifically mature bodies of goats. Do you mean mature microRNAs? If so please clarify.
Answer 2:The tags were compared with miRNA precursors and the mature bodies by utilizing the miRBase database (release 21) (http://www.MiRbase.org/) to obtain known miRNAs.
Question 3:L228: Results:Please include more details about the follicle groups and descriptive characteristics on the follicles utilized for this study. It is important for the reader to understand the differences present between the GF groups. For example is the GF_1 group >5m while GF_6 <3 or something similar? What was the median follicle size for each of the comparison groups? The characteristics of the follicles used are results and should be included.
Answer 3:In view of the above problems, we have modified the description of follicles in the results of the paper, but more detailed information.
According to the previous study results on determination of follicular development in Dazu black goat by B-ultrasound technology from our lab, the diameter of Graafian follicle was defined as greater than 4mm [27]. Therefore, the diameter length of follicles larger than 4 mm and smaller than 4 mm was divided into GF_T and GF_C, respectively. The follicle groups GF_1 to GF_6 were defined by decreasing of their diameter.
